The high diversity of gametogenic pathways in amphispermic water frog hybrids from Eastern Ukraine

Pustovalova Eleonora 1 2 3
Choleva Lukaš 1 2
Shabanov Dmytro 3
Dedukh Dmitrij dmitrijdedukh@gmail.com 1
1 Laboratory of Fish Genetics, Institute of Animal Physiology and Genetics of the CAS, v.v.i. , Libechov , Czech Republic
2 Department of Biology and Ecology, Faculty of Science, University of Ostrava , Ostrava , Czech Republic
3 Laboratory of Amphibian Population Ecology, Department of Zoology and Animal Ecology, School of Biology, V. N. Karazin Kharkiv National University , Kharkiv , Ukraine
Fernandes Carlos Eurico
Electronic publication date: 2022 Aug 23
Publication date: 2022
Volume: 10
Electronic Location ID: e13957
Received 2022 Jun 23; Accepted 2022 Aug 6
Copyright: ©2022 Pustovalova et al.
Copyright year: 2022
Copyright holder: Pustovalova et al.
License: This is an open access article distributed under the terms of the Creative Commons Attribution License, which permits unrestricted use, distribution, reproduction and adaptation in any medium and for any purpose provided that it is properly attributed. For attribution, the original author(s), title, publication source (PeerJ) and either DOI or URL of the article must be cited.
License URL: https://creativecommons.org/licenses/by/4.0/

Keywords: Gametogenesis, Spermatid, Meiosis, Pelophylax, Amphispermy, FISH, Bivalents, Hybridogenesis

Funding: Czech Science Foundation 21-25185S GA19-24559S IAPG, AS CR, v.v.i Institutional Research Concept RVO67985904 Lukáš Choleva, Eleonora Pustovalova, Dmitrij Dedukh were funded by the Czech Science Foundation (Grantová Agentura Cbreveeské Republiky; project no. 21-25185S), and IAPG, AS CR, v.v.i Institutional Research Concept RVO67985904 (Ústav živočišnéfyziologie a genetiky Akademie věd České republiky, v.v.i). Lukáš Choleva, Dmitrij Dedukh were funded by the Czech Science Foundation (grant no. GA19-24559S). The funders had no role in study design, data collection and analysis, decision to publish, or preparation of the manuscript.

==============================
Interspecific hybridization can disrupt canonical gametogenic pathways, leading to the emergence of clonal and hemiclonal organisms. Such gametogenic alterations usually include genome endoreplication and/or premeiotic elimination of one of the parental genomes. The hybrid frog Pelophylax esculentus exploits genome endoreplication and genome elimination to produce haploid gametes with chromosomes of only one parental species. To reproduce, hybrids coexist with one of the parental species and form specific population systems. Here, we investigated the mechanism of spermatogenesis in diploid P. esculentus from sympatric populations of P. ridibundus using fluorescent in situ hybridization. We found that the genome composition and ploidy of germ cells, meiotic cells, and spermatids vary among P. esculentus individuals. The spermatogenic patterns observed in various hybrid males suggest the occurrence of at least six diverse germ cell populations, each with a specific premeiotic genome elimination and endoreplication pathway. Besides co-occurring aberrant cells detected during meiosis and gamete aneuploidy, alterations in genome duplication and endoreplication have led to either haploid or diploid sperm production. Diploid P. esculentus males from mixed populations of P. ridibundus rarely follow classical hybridogenesis. Instead, hybrid males simultaneously produce gametes with different genome compositions and ploidy levels. The persistence of the studied mixed populations highly relies on gametes containing a genome of the other parental species, P. lessonae.

Introduction

Meiosis is a conserved process for all eukaryotic organisms and represents a hallmark of sexual reproduction (Lenormand et al., 2016). Chromosome conjugation during meiosis relies on sufficient homology between chromosomes (McKee, 2004), whereas insufficient pairing may lead to meiotic abruption and formation of aneuploid gametes. These mechanisms keep taxa prezygotically reproductively isolated (Zong & Fan, 1989; Borodin et al., 1988; Ishishita et al., 2015; Torgasheva & Borodin, 2016; Dedukh et al., 2020). Interspecific hybridization has both positive (Mallet, 2007; Abbot et al., 2013) and negative impacts (Arnold & Hodges, 1995; Rieseberg, 2001; Coyne & Orr, 2004) and plays a key role in evolution. One of the outcomes of hybridization is the creation of individuals with clonal and hemiclonal reproductive modes (Dawley & Bogart, 1989; Schön, Martens & Van Dijk, 2009; Neaves & Baumann, 2011; Stöck et al., 2021). Hybrid clonal animals form gametes with a chromosomal composition identical to that of their somatic cells (Dawley & Bogart, 1989; Schön, Martens & Van Dijk, 2009; Neaves & Baumann, 2011; Stöck et al., 2021). Hybrid hemiclonal animals produce unrecombined haploid gametes that require fertilization to restore diploid chromosomal sets in their offspring (Dawley & Bogart, 1989; Schön, Martens & Van Dijk, 2009; Stöck et al., 2021; Dedukh & Krasikova, 2021). A switch to asexual reproduction requires significant modifications to gametogenesis, rescuing hybrids from sterility, and the creation of alternative pathways for successful reproduction. Thus, our understanding of reproductive ability and evolutionary potential of hybridization lies in our understanding of hybrid gametogenesis.

Hemiclonal reproduction, also known as hybridogenesis, has been found in European water frogs of the genus Pelophylax (Tunner, 1974). This animal system includes two parental species: P. lessonae (Camerano, 1882) (LL genotype) and P. ridibundus (Pallas, 1771) (RR genotype), and their hybrid P. esculentus (Linnaeus, 1758). Hybrids can be represented in diploid (RL) and triploid (LLR, LRR) forms (Günther, Uzzell & Berger, 1979; Berger, 1983). The classical model of hybridogenetic reproduction states that one parental genome is eliminated during gametogenesis while the other is duplicated and transmitted to gametes, which appear to be clonal (Tunner, 1973; Tunner & Heppich, 1981; Tunner & Heppich-Tunner, 1991; Chmielewska et al., 2018; Doležálková-Kaštánková et al., 2021). Triploid hybrids usually eliminate a genome present in one copy, whereas the genome present in two copies enters meiosis and forms recombinant gametes (Günther, Uzzell & Berger, 1979; Graf & Polls-Pelaz, 1989; Ogielska, 1994; Plötner, 2005; Christiansen & Reyer, 2009; Dedukh et al., 2015; Dufresnes & Mazepa, 2020; Dedukh et al., 2020; Dedukh & Krasikova, 2021). However, the detailed principles of genome elimination and duplication during hybrid gametogenesis remain unknown.

Hybridogenetic gametogenesis makes hybrids dependent on parental species and leads to the formation of population systems where hybrids coexist with one or both parental species, or for all-hybrid populations with various ploidy and genomic compositions (Graf & Polls-Pelaz, 1989; Plötner, 2005; Christiansen & Reyer, 2009). In most of the distribution range, P. esculentus coexists with P. lessonae, creating the L-E system (Graf & Polls-Pelaz, 1989; Plötner, 2005; Pruvost, Hoffmann & Reyer, 2013; Svinin et al., 2013; Svinin et al., 2021; Hoffman et al., 2015; Dufresnes & Mazepa, 2020). Here, hybrids have a typical hemiclonal gametogenesis with preferential elimination of the P. lessonae genome, followed by the transmission of P. ridibundus genome to gametes (Günther, 1983; Bucci et al., 1990; Pruvost, Hoffmann & Reyer, 2013; Dedukh et al., 2019; Svinin et al., 2021). The R-E system forms hybrids mixed in populations with P. ridibundus. P. esculentus from this system is specific to significant alterations in gametogenic pathways, resulting in decreased fertility and increased numbers of aneuploid gametes (Uzzell, Günther & Berger, 1977; Günther, 1983; Vinogradov et al., 1991; Borkin et al., 2004; Ragghianti et al., 2007; Doležálková et al., 2016; Dedukh et al., 2015; Dedukh et al., 2017; Biriuk et al., 2016). Studies of geographic variation showed that in Central Europe (Doležálková et al., 2016; Doležálková-Kaštánková et al., 2018; Doležálková-Kaštánková et al., 2021), P. esculentus is present only in a male sex, and both sexes of P. ridibundus coexist in Eastern Europe. P. esculentus syntopic with P. ridibundus is present in both sexes and at two ploidy levels (RL, RRL, and LLR) (Borkin et al., 2004; Shabanov et al., 2020).

Previous studies from Eastern Ukraine have shown that hybrid females frequently produce haploid gametes with the R genome and diploid gametes with the RL genome, whereas gametes with L genomes have never been detected (Dedukh et al., 2015; Dedukh et al., 2017). Additionally, diploid hybrid males usually simultaneously produce a mixture of gametes with the L and R genomes. This phenomenon, called hybrid amphispermy (Vinogradov et al., 1991), includes the simultaneous formation of L and R sperms, and was first observed in Central Europe (Vinogradov et al., 1991; Doležálková et al., 2016). Vinogradov et al. (1991) suggested the existence of at least two germ cell populations that can eliminate either P. ridibundus or P. lessonae genome during amphispermic reproduction. An alternative hypothesis proposed the absence of premeiotic genome elimination and a different separation of the L and R genomes in the first meiotic division (Doležálková et al., 2016).

In the current study, we analyzed hybridogenetic gametogenesis in Eastern Europe. Using fluorescent in situ hybridization (FISH) with probe RrS1 specific to centromeric regions of P. ridibundus chromosomes, we identified the genomes of P. ridibundus during metaphase of meiosis I, spermatids, and mitotic spreads on chromosomal spreads from hybrid male gonads. Combining these data, we tested (i) whether amphispermy is widespread gametogenesis in hybrid males over R-E systems from Eastern Ukraine. Further, we tested (ii) whether premeiotic genome elimination of both L and R genomes occurs in different gonial cells of amphispermic males, or not.

Materials and methods

Samples

Sampling was conducted in Kharkiv Oblast, Eastern Ukraine, during 2016–2019. We collected six adult P. esculentus males from the Mozh River (49.749167; 36.162778), five males from the Iskiv water body (49.627778; 36.282778), and one male from the Udy River (49.968333; 36.136944) (Fig. S1). These geographically isolated population systems are characterized by the coexistence of di- and triploid hybrids of both sexes, represented by LR, LLR, and LRR genotypes, and P. ridibundus of both sexes. Animals were caught at night using a torch. All specimens were collected outside of the protected areas within Eastern Ukraine and therefore, no specific permissions were required. All animal manipulations were performed according to national and international guidelines. Standard techniques for capture, tissue sampling, and euthanasia were used to minimize animal suffering. Before euthanasia, each individual was anesthetized by submersion in ethyl ethanoate (ETAC). All procedures were approved by the Committee on Bioethics of the V. N. Karazin Kharkiv National University (minutes No 4, 21.04.2016). The previous species and ploidy identification were determined by a complex of morphological features and Ag-staining (Birstein, 1984) with some modification and further confirmed within the preparation of somatic tissue chromosomes followed by fluorescent in situ hybridization (FISH) with species-specificity (Ragghianti et al., 1995; Dedukh et al., 2015; Dedukh et al., 2017).

Preparation of mitotic and meiotic chromosomes

Before euthanasia in ETAC, each frog was injected with 0.05% colchicine for 12 h. The intestines and testes were dissected, cleaned, and treated hypothonically (0.07M KCl) for 20 min. The tissues were transferred to Carnoy’s fixative (3:1 methanol: glacial acetic acid), and the solution was changed thrice. To prepare chromosomal spreads, the tissue fragments were transferred to 70% acetic acid solution for maceration in a suspension of cells and dropped onto slides pre-heated to 60 °C (Biriuk et al., 2016). The chromosomal and cell nuclei spreads were dried on a heating table at 60 °C for 1 h.

Fluorescent in situ hybridization

Male gametogenesis was further analyzed using the FISH method on mitotic and meiotic chromosomes, following Dedukh et al. (2015) and Dedukh et al. (2017). The slides were treated with RNAse (100–200 µg/ml) for 1 h and pepsin D (0.005%, diluted in 0.01 N HCl) for 3 min. The probe was labelled with biotin l from the genomic DNA of P. ridibundus by PCR using the following primers to RrS1 centromeric repeat: 5′-AAGCCGATTTTAGACAAGATTGC- 3′; 5′-GGCCTTTGGTTACCAAATGC- 3′. The probe was added to the hybridization mixture (50% formamide, 1 µl 2xSSC and tRNA, 10% dextran sulphate, 1.5 µl labelled probe). Slides containing mitotic and meiotic chromosomes were denatured at 77 °C for 3 min and incubated at room temperature for 12–18 h. The slides were then washed thrice in 0.2xSSC at 60 °C. Biotin was detected using avidin conjugated with the fluorochrome Alexa 488 or Cy3. After washing in 4xSSC slides, they were dehydrated in an ethanol series, air-dried, and mounted in DABCO antifade solution containing 1 µg/ml DAPI.

Image processing

Mitotic and meiotic chromosomes were inspected after FISH using Provis AX70 Olympus microscopes and Leica DM 2000 equipped with standard fluorescence filter sets. Microphotographs of chromosomes were captured with a CCD camera (DP30W Olympus) using Olympus Acquisition Software and a Leica DFC3000 G camera using Leica LASX Software. Microphotographs were adjusted and arranged in the Adobe Photoshop CS6 software. FISH-based mapping of RrS1 pericentromeric repeats visualizes the centromeric regions of P. ridibundus chromosomes (Ragghianti et al., 1995), but cannot identify P. lessonae genome during interphase. The analysis allowed us to discriminate different gametogenic stages, as we identified the presence of P. ridibundus genome in mitotic (from both somatic and germ cells) and meiotic chromosome plates as well as in the nuclei of somatic and germ cells and spermatids (Table S1). Interphase cells and spermatids with 5–13 signals were discriminated as cells with P. ridibundus genome. Among these signals five were usually bright and clearly distinguishable while remaining eight signals were either weak or absent. Cells with 1–4 signals were not taken into account. Five signals observed in interphase cells and spermatids corresponded to the haploid P. ridibundus chromosomal set, where we observed five bright signals on all large chromosomes and one small chromosome while signals on the other chromosomes were either weak or absent. Ragghianti et al.; Ragghianti et al. (1995 and 2007) observed six signals in interphase cells of diploid hybrids. Interestingly, Dedukh et al.; Dedukh et al. (2019 and 2020) detected 13 signals in a haploid set of P. ridibundus chromosomes, while they also found a difference in the signal intensity. A signal variation and polymorphism of the studied pericentromeric repeat may explain technical differences in laboratory protocol used, the source of genomic DNA used for probe preparation or the interpopulation polymorphism.

Results

The two geographically isolated populations of P. esculentus were characterized by the coexistence of diploid and polyploid hybrids. Here, we used FISH with the RrS1 probe to identify the genome composition of interphase nuclei, spermatids, and meiotic and mitotic chromosomal plates obtained from the testes of 11 diploid P. esculentus males. The hybrid testes were round in shape without any visible anomalies. In nine males, the left testis was larger than the right (left mean 5.8 mm; right mean 4.1 mm) and two males had testes of equal sizes (frogs’ ID: 19I-60, 19I-62) (Table S2). Testes size difference is common in P. esculentus and might be accompanied by decreased fertility (Berger, 1970; Ogielska & Bartmańska, 1999). Data from a single male from the Udy River (17U−4.2) were insufficient to evaluate hybrid gametogenesis in this locality. Raw data on the number of each type of gametes produced by this male are presented in Table S1.

Gametogenesis in diploid hybrid males in Mozh River

Analysis of 436 interphase nuclei from four diploid hybrid males (17T-5, 17T-10, 18T-8, 18T-7) showed the presence of interphase nuclei with 3–18 signals (Figs. 1D, 1E, 1G, 1H and 1J) along with interphase nuclei without signals (Fig. 1H). Interphase nuclei without signals were those with exclusive content of P. lessonae chromosomes. Nuclei with 5–13 signals contained at least a haploid set of P. ridibundus chromosomes, whereas nuclei with more than 13 signals contained an aneuploid or diploid chromosomal set of P. ridibundus. The analysis of 79 metaphase plates during mitosis showed 0–24 signals, among which most metaphase plates had 12–13 signals (Fig. 1E). These results fit well with the interphase nuclei analysis, suggesting at least three cell populations: cells with 26 P. lessonae chromosomes, cells with 13 P. ridibundus and 13 P. lessonae chromosomes, and cells with 26 P. ridibundus chromosomes. Distinguishing germ cells from somatic cells is difficult. However, as genome elimination and endoreplication occur only in germ cells, we considered cells with P. lessonae chromosomes as germ cells. During meiosis I, we observed spermatocytes with 13 bivalents of P. ridibundus and spermatocytes with 13 bivalents of P. lessonae in all four males analyzed (Figs. 1F and 1G). In two of these males (18T-7, 17T-10), bivalents with P. ridibundus chromosomes dominated (87% and 77%). During meiosis II, we detected spermatocytes with 13 univalents of P. ridibundus chromosomes (Fig. 1H) and 13 univalents of P. lessonae chromosomes (Fig. 1I). Additionally, we observed many cells with aberrant pairing in all analyzed males. The observed hybrids potentially eliminated different genomes in different cells premeiotically, or had some problems with selective elimination. We detected spermatids in which the signal of P. ridibundus probe varied from 0 to 12, suggesting the presence of spermatids in P. lessonae and P. ridibundus genomes (Figs. 1D and 1J). These males transmitted two parental genomes in their cells simultaneously, i.e., they were amphigametic.

Figure 1 Identification of ploidy level and genome composition of gonocytes, spermatocytes, and spermatids from P. esculentus males collected from the Mozh river basin.

FISH with RrS1 probe helps distinguish pericentromeric regions only of P. ridibundus chromosomes (indicated by thin arrows). (A–C) Somatic cells (C), spermatids (B, C), and spermatocytes in meiosis I (A) and II (B) had only P. ridibundus chromosomes suggesting the presence of premeiotic genome elimination of P. lessonae genome and endoreplication of P. ridibundus genome. (D–J) Germ line cells (gonocytes, spermatocytes, and spermatids) with different ploidies suggesting the presence of premeiotic elimination and endoreplication of different genomes in various cell lines. Interphase cells (indicated by thick arrows) with a haploid set of P. ridibundus chromosomes (D, E, G, H, J) and with P. lessonae chromosomes (I). Mitotic metaphase cell with 13 P. ridibundus chromosomes and 13 P. lessonae chromosomes (E). Meiotic metaphase I with 13 bivalents of P. ridibundus (D, F, J) and 13 bivalents of P. lessonae (G). Meiotic metaphase II with 13 univalents of P. ridibundus (H) and 13 univalents of P. lessonae (I). Spermatids (indicated by arrowheads) with haploid set of P. ridibundus chromosomes (D, J) and P. lessonae (D, J). Scale bar = 10 µm.

Fifty-four examined interphase cells of one male (18-T6) had at least five signals, indicating the presence of the haploid P. ridibundus genome (Fig. 1C). The analysis of 14 mitotic chromosomal plates showed 8 plates with 26 chromosomes, of which 13 belonged to P. ridibundus and 13 to P. lessonae, the other six mitotic chromosomal plates were aneuploid. During the analysis of 32 metaphases of meiosis I, we detected 13 bivalents of P. ridibundus (Fig. 1A). We also detected five metaphases of meiosis II with 13 univalents of P. ridibundus (Fig. 1B). In addition, 24 aneuploid chromosomal plates (Fig. 1C) were observed. The analyzed spermatids (n = 48) exclusively exhibited the presence of P. ridibundus chromosomes. We suggest that during gametogenesis in this male, the genome of P. lessonae was premeiotically eliminated, followed by endoreplication of the P. ridibundus genome.

In one individual (17T-8), we observed interphase nuclei with 3–26 signals (Figs. 2A, 2B and 2D). Haploid P. ridibundus genome was suggested in cells with 5-13 signals; diploid P. ridibundus genome was suggested in cells with 15–26 signals. The analysis of 14 mitotic chromosomal plates from this individual showed 3 mitotic chromosomal plates with approximately 52 chromosomes, including chromosomes exclusive to P. ridibundus (Fig. 2B) and chromosomes exclusive to P. lessonae (Fig. 2C). In 8 metaphase plates, we observed 26 chromosomes exclusive to P. ridibundus (Fig. 2D) as well as both P. ridibundus and P. lessonae chromosomes (Fig. 2I). In meiosis I, we detected chromosomal plates with 13 tetravalents of P. ridibundus and metaphase plates with 13 tetravalents of P. lessonae (Fig. 2G) (23% of the total amount). One of the genomes was eliminated to form spermatocytes with genome-specific tetravalents, whereas the other underwent two rounds of genome endoreplication. We also found metaphase plates of meiosis I with approximately 13 tetravalents, including 26 chromosomes of P. ridibundus and 26 chromosomes of P. lessonae (Figs. 2C and 2F). Spermatids of this male had 3–19 signals, suggesting the presence of two P. ridibundus genomes at least in some spermatids (Figs. 2F–2H). This pattern also supports the amphigametic production.

Figure 2 Identification of ploidy level and genome composition of gonocytes, spermatocytes and spermatids from particular P. esculentus male producing diploid spermatids collected from the Mozh river basin.

Interphase cell nuclei (indicated by thick arrows) with diploid P. ridibundus chromosomal set (A, D). Mitotic metaphases with 26 P. ridibundus chromosomes (D), approximately 47 P. ridibundus chromosomes (B), approximately 40 P. lessonae chromosomes (C) and 13 P. ridibundus and 13 P. lessonae chromosomes. Meiotic metaphase I with 13 P. ridibundus bivalents (A, H), approximately 12 tetravalents (or mixture of bivalents and tetravalents) with chromosomes exclusive to P. ridibundus (E), and with approximately 11 tetravalents with chromosomes exclusive to P. lessonae (G). Meiotic metaphase I with a mixture of approximately nine P. lessonae tetravalents and four P. lessonae bivalents as well as four P. ridibundus tetravalents and four P. ridibundus bivalents. Spermatids (shown by arrowheads) with at least five P. ridibundus chromosomes (designated as haploid P. ridibundus genome) (B, H), with only P. lessonae chromosomes (designated as haploid or diploid P. lessonae genome) and at least 14 P. ridibundus chromosomes and at least 17 P. ridibundus chromosomes (designated as diploid P. ridibundus genome) (F, H). P. ridibundus chromosomes identified using FISH-based detection of pericentromeric RrS1 repeats (indicated by thin arrows). Scale bar = 10 µm.

Gametogenesis in diploid hybrid males in Iskiv pond

Analysis of interphase nuclei of one male (19I-60) revealed both interphase cells without signals and those with RrS1 signals (Fig. S2J). Some cells had, therefore, chromosomes exclusive to P. lessonae, and some cells had at least one haploid genome of P. ridibundus. Mitotic metaphase plates of this individual were represented by 26 chromosomes, with 13 P. ridibundus chromosomes, 13 P. lessonae chromosomes, and 26 chromosomes exclusive to P. ridibundus (Fig. S2J). Our metaphase inspection of meiosis I clearly distinguished 13 P. ridibundus bivalents (Figs. S2K–S2L). To form such spermatocytes, P. lessonae genome must have been premeiotically eliminated, whereas P. ridibundus genome was endoreplicated. Additional aneuploid cells (n = 30) suggest aberrant genome elimination and endoreplication. The analysis of spermatids (n = 29) revealed that most spermatids had P. lessonae genome, and only a few spermatids had P. ridibundus genome (Fig. S2L). Though we observed both interphase nuclei and spermatids exclusively in the P. lessonae genome, we did not detect meiotic plates with P. lessonae bivalents. Therefore, we suggest that spermatocytes with P. lessonae must be present in this individual, i.e., the individual was amphispermic with the prevalence of L-gametes.

The analysis of interphase nuclei (n = 307) from two males (19I-62 and 18I-90) showed some interphase nuclei only in P. lessonae chromosomes and others in P. ridibundus chromosomes (Fig. S2A–S2C). During the analysis of mitotic metaphases (n = 44), we detected metaphase plates with 26 chromosomes, including 13 P. ridibundus and 13 P. lessonae chromosomes (Fig. S2B). Most spermatocytes had 13 bivalents of P. ridibundus (Fig. S2C) while only a few spermatocytes had 13 P. lessonae bivalents. We detected 58 aneuploid chromosome plates in both males (Fig. S2D). In meiosis II, we observed spermatocytes with 13 univalent P. ridibundus and 13 univalent P. lessonae (Fig. S2A). In spermatids (n = 114), we found those with P. ridibundus chromosomes and exclusive P. lessonae chromosomes (Fig. S2B), supporting the pattern of amphigametic production.

Analysis of interphase nuclei (n=110) in two other males (18I-91 and 19I-61) revealed nuclei exclusively with P. lessonae chromosomes and nuclei with P. ridibundus chromosomes (Figs. S2E–S2G, S2I). During the analysis of mitotic metaphases (n=13) obtained from the other male (19I-61), we found metaphase plates with 26 chromosomes, among which 13 chromosomes were from P. lessonae and 13 were from P. ridibundus (Fig. S2E), while mitotic chromosomal plates were not detected in one of the males (18I-91). Both males simultaneously produced spermatocytes with 13 P. ridibundus bivalents (Fig. S2F) and 13 P. lessonae bivalents. During meiosis II, we detected spermatocytes with 13 P. lessonae univalents (Figs. S2H and S2I) and with 13 P. ridibundus univalents (Fig. S2G). In spermatids, the number of signals was varied from 0 to 13. Spermatids with no signal were considered as bearing P. lessonae genome (Fig. S2I); spermatids with 5-13 were considered as bearing P. ridibundus genome (Figs. S2F–S2H). These two males (18I-91, 19I-61) potentially eliminated different genomes in different cells premeiotically and transmitted the two genomes in their cells, thus being amphigametic.

Discussion

Diverse spermatogenesis in diploid hybrids

Our study of hybrid P. esculentus males from Eastern Ukrainian populations revealed diverse gamete formation (Fig. 3, Fig. S3, Table S1). Nine out of eleven males simultaneously produced two types of haploid gametes with parental chromosomes (amphispermic male, Fig. 4, Pathway III), one with P. lessonae genome and one with P. ridibundus genome, free of recombination and crossover between the genomes of parental species. A single male represented the second type of spermatogenesis-producing spermatid with P. ridibundus genome only (Fig. 3B, Table S1). We also found a male suspected to form diploid sperm based on sperm analysis and tetravalent observations during meiosis, which corresponded to the third type of spermatogenesis (Figs. 3B and 3D). The simultaneous production of fertile gametes with P. lessonae and P. ridibundus genomes (amphispermy) was determined using DNA flow cytometry in the Iskiv pond population (Biriuk et al., 2016) and from artificial crosses in the Mozh River (Mazepa et al., 2018). By analyzing the process of gametogenesis in detail, we provide clear pathways on the mechanisms of the origins of diverse gametes in these tetrapod animals.

Figure 3 Relative number of normal and aneuploid chromosomal plates during mitosis (A, C) and meiosis (B, D) from hybrid frogs collected from the R-E system of the Mozh river (A, B) and Iskiv pond (C, D).

R, genome of P. ridibundus; L, genome of P. lessonae; aneuploidy, number of chromosomes more or less 13 bivalents or univalents.

Figure 4 Suggested gametogenic pathways in sexual species and hybrid males from studied R-E systems.

Pathway I: Genome elimination and endoreplication (‘classical’ hybridogenesis). During classical genome elimination, one of the parental genomes is eliminated before meiosis, whereas the other is endoreplicated, allowing the restoration of the diploid chromosome set. These cells undergo meiotic division with 13 bivalents during meiosis I and 13 bivalents during meiosis II. Subsequent spermatids bear the genomes of only one parental species (P. ridibundus or P. lessonae). Pathway II: Genome elimination of one of the parental species (P. ridibundus or P. lessonae) during meiosis. This type of gamete formation also involves the elimination of only one parental genome. However, it occurs directly during meiosis. After meiotic divisions I (13 bivalent stages) and II (13 univalent stages), spermatids bear the endoreplicated genome. Pathway III: The genomes of different parental species were eliminated from different germline populations. Therefore, some gonocytes bear only P. ridibundus chromosomes, whereas some cells have P. lessonae chromosomes only. Germ cells with both parental genomes duplicated and formed two types of parental species bivalents (2n = 26). After meiosis II, the spermatids were from both parental species (P. ridibundus and P. lessonae). Pathway IV: Diploid sperm formation. Two rounds of endoreduplication of one parental species genome resulted in the formation of tetravalents, bearing four sets of P. ridibundus or P. lessonae genomes in meiosis I. Such cells, which have undergone meiosis II, bear a double chromosome set (RR, LL, or even RL). Pathway V: Abnormal meiosis. Due to disruptions during the elimination of P. ridibundus or P. lessonae genome, there are no vital spermatids, so the individual is sterile.

Inspecting meiosis, we revealed spermatocytes with 13 univalents or bivalents of P. ridibundus (39% for Mozh, 47% for Iskiv, 43% for both) as well as 13 univalents or bivalents of P. lessonae (32% for Mozh, 20% for Iskiv, 26% for both) (Fig. S3A). Interphase nuclei and mitotic chromosomes from testis cell suspensions often bear either P. ridibundus or P. lessonae chromosomes (Figs. 3A and 3C). The methodology used cannot distinguish whether interphase nuclei and metaphase chromosomes belong to germ cells or somatic cells. However, as genome elimination and endoreplication occur only in the germ cells, we considered the observed cells as germ cells. As we detected germ cells and spermatocytes bearing only P. ridibundus or P. lessonae chromosomes, we suggest that genome elimination and endoreplication occurred in germ cells before meiosis (Fig. 4, Way III). A phenomenon of premeiotic genome elimination has been described earlier in water frog hybrids during tadpole development and causes the classical formation of a single gamete type (Tunner & Heppich-Tunner, 1991; Ogielska, 1994; Dedukh et al., 2017; Dedukh et al., 2019; Dedukh et al., 2020; Chmielewska et al., 2018). The presence of cells with only P. ridibundus and P. lessonae genomes indicated the existence of at least two cell population types eliminating different parental genomes, even in a single individual, as proposed by Vinogradov et al. (1991). Comparative genomic hybridization on Central-European amphispermic males has revealed meiotic metaphase I with univalent and bivalent-like configurations, including bivalent-like configurations between the two parental genomes (Doležálková et al., 2016). Based on these observations, Doležálková et al. proposed a hypothesis in which premeiotic elimination would be absent in these cases, followed by segregation of P. ridibundus and P. lessonae chromosomes during meiosis I. Diploid hybrid males from Eastern Europe likely do not use this hypothetical strategy, as evidenced by our observation of premeiotic genome elimination followed by genome duplication in different germ cell populations (Fig. 4). However, it should be noted that bivalent-like configurations between the two different parental genomes were not observed in our males. The presence of aneuploid cells during meiosis (on average 25% for Mozh, 33% for Iskiv, 29% for both) indicates problems with genome elimination and/or endoreplication (Fig. 4, Way V). Aneuploid meiocytes and meiocytes with unusual pairings were detected earlier in both hybrid females (Dedukh et al., 2015; Dedukh et al., 2017) and males (Biriuk et al., 2016) from the same locality and generally in various population types (Heppich, Tunner & Greilhuber, 1982; Bucci et al., 1990; Christiansen et al., 2005; Christiansen, 2009; Christiansen & Reyer, 2009; Dedukh et al., 2019). It should be noted that aberrations were highly numerous in hybrid frogs from a mixed population of P. ridibundus, suggesting difficulties in genome elimination and duplication during hybrid gametogenesis (Uzzell, Günther & Berger, 1977; Ragghianti et al., 2007; Doležálková et al., 2016; Dedukh et al., 2015; Dedukh et al., 2017; Biriuk et al., 2016).

A single hybrid male produced spermatocytes with 13 tetravalents of P. ridibundus and 13 tetravalents of P. lessonae, indicating that it underwent an additional round of genome duplication (Fig. 3B). To form spermatocytes with 13 tetravalents of P. ridibundus, the cells must first eliminate P. lessonae chromosomes, followed by two rounds of duplication of P. ridibundus chromosomes, and vice versa for P. lessonae tetravalents (Fig. 4, Way IV). Additional detection of spermatocytes with 13 tetravalents during meiosis I with both genomes of the parental species suggests the absence of genome elimination and two rounds of genome endoreplication. Interphase cells with 26 P. ridibundus chromosomes (Fig. 2A) resembled the results obtained for the diploid hybrid males with metaphase plates and tetravalents (Ragghianti et al., 2007). Similar observations were made by Dedukh et al. (2015) during lampbrush chromosome analysis, where the authors found one hybrid female with 26 P. ridibundus bivalents. In addition, such a pattern supports the presence of two rounds of genome endoreplication preceding meiosis after the elimination of one of the parental genomes. Chromosomal plates with tetravalents are typically formed in autopolyploid frogs of the Pleuroderma genus (Salas et al., 2014). Nevertheless, in these species, bi-, tetra-, and octavalents were also detected among metaphase plates, suggesting some pairing inaccuracies (Salas et al., 2014). Bi & Bogart (2010) showed the presence of quadrivalents (the same as tetravalents) in Ambystoma hybrid females by investigating lampbrush chromosomes, suggesting occasional synapses between homologous chromosomal regions. Nevertheless, such oocytes are a rare phenomenon in Ambystoma (Bi & Bogart, 2010), while in water frogs, we provide frequent observations with numbers of spermatocytes with tetravalents varying in their genome composition. We hypothesized that these cells could proceed through meiosis and form diploid sperm with the LL, RL, and RR genomes (Fig. 4, Way IV). Such gametes may lead to the emergence of triploid frogs (approximately 5%) observed in the Mozh Basin (Drohvalenko et al., 2022). However, the fertilization success of diploid sperms to compete with haploid sperms requires further investigation.

As not only hybrid males but hybrid females (Dedukh et al., 2015; Dedukh et al., 2017; Christiansen & Reyer, 2009; Christiansen & Reyer, 2009; Pruvost, Hoffmann & Reyer, 2013) can also produce gametes of both parental species, Dubey et al. (2019) called this phenomenon as ‘amphigamy’. However, this term has following interpretations according to Rieger, Michaelis & Green (1991): (1) the fusion of two sex cells and the formation of conjugated pairs of nuclei (dikaryophase). If amphigamy immediately follows karyogamy, the process is referred to as amphimixis (Renner, 1916); and (2) the normal fertilization process (Battaglia, 1947). Therefore, we considered correcting the term to ‘amphigameticity’ to indicate the ability of interspecific hybrid males and females to produce gametes of both parental species.

The gain and loss during diverse gamete formation

To establish successful hemiclonal genome propagation, hybrid organisms must adapt gametogenesis accordingly. The F1 hybrids of P. ridibundus and P. lessonae showed premeiotic genome elimination and endoreplication, rescuing their fertility (Tunner & Heppich-Tunner, 1991; Dedukh et al., 2019). However, premeiotic genome elimination and endoreplication do not occur in all populations of germ cells, causing unusual pairing in meiosis and abruption of gamete formation, thereby decreasing fertility in otherwise vital individuals (Vorburger, 2001; Dedukh et al., 2015; Dedukh et al., 2019; Dedukh et al., 2020; Doležálková et al., 2016). Reported cases of genome elimination and/or endoreplication failure cause the formation of aneuploid cells during mitosis and meiosis (Fig. 3, Fig. S3). Not all changes in genome elimination and endoreplication machinery harm the reproduction of hybrid frogs. At least one hybrid male from Eastern Ukraine potentially produced diploid spermatozoa with LL, RL, and RR genomes. The formation of diploid gametes is crucial for the emergence of triploid hybrids in some population systems (Tunner & Heppich-Tunner, 1992; Brychta & Tunner, 1994; Rybacki & Berger, 2001; Mikulícek & Kotlík, 2001; Pruvost et al., 2015).

We stress that hybrids have an additional challenge in the selective elimination of P. ridibundus genome. During the initial crossing of P. ridibundus and P. lessonae, hybrids usually transmit the P. ridibundus genome and eliminate P. lessonae (Berger, 1971; Dedukh et al., 2019). Subsequent backcrosses of diploid hybrids with P. lessonae individuals ensure the maintenance of hybrids and lead to the formation of a mixed population of hybrids and P. lessonae (Berger, 1971; Günther, 1983; Christiansen & Reyer, 2009). Hybridogenetic reproduction of hybrid frogs in this population type is characterized by stable propagation of P. ridibundus genome with relatively rare aberrations in genome elimination and endoreplication (Berger, 1971; Graf & Müller, 1979; Pruvost, Hoffmann & Reyer, 2013; Dedukh et al., 2019). Surprisingly, a growing number of evidence shows that also hybrid frogs in a mixed population with P. ridibundus produced mostly R gametes and/or L gametes (this study; Uzzell, Günther & Berger, 1977; Graf & Polls-Pelaz, 1989; Vinogradov et al., 1991; Dedukh et al., 2015; Dedukh et al., 2017; Biriuk et al., 2016, for the exceptions see Doležálková-Kaštánková et al., 2021), although the L gametes are the crucial cells for the hybrid’s persistence (Fig. S3C). As haploid gametes with P. ridibundus genome would not lead to hybrid progeny when coexisting with P. ridibundus, it is clear that these hybrids have to under absence of P. lessonae produce fertile P. lessonae gametes to perpetuate themselves. Obvious difficulties in forming gametes with P. lessonae genome may explain why mixed populations of hybrids and P. ridibundus are rare over continental Europe compared to mixed hybrid populations with P. lessonae (Uzzell, Günther & Berger, 1977; Graf & Polls-Pelaz, 1989; Plötner, 2005). For example, the evolutionary origin of P. ridibundus–P. esculentus male populations in Central Europe seems to be rare event in the past time, as clonally inherited lessonae genomes share their ancestors (Doležálková et al., 2016; Doležálková-Kaštánková et al., 2018; Doležálková-Kaštánková et al., 2021).

In this light of the evidence, diploid hybrid males persisting within the R-E system in Eastern Europe in high numbers over decades of observation (Borkin et al., 2004; Shabanov et al., 2020) remains unclear. As hybrid males produce mainly a mixture of R and L genomes (Fig. 3, Fig. S3), while female and co-occurring triploid hybrids with the RRL genotype produce R and RL gametes, the proportion of hybrids that received P. lessonae gametes is expected to be lower than observed. Moreover, long-term clonal propagation of the genome may theoretically lead to the accumulation of deleterious mutations, thus decreasing the survival of hybrids (Tunner & Heppich-Tunner, 1991; Christiansen et al., 2005; Christiansen, 2009; Dubey et al., 2019). The maintenance of these hybrid males may explain different competition rates between P. esculentus and P. ridibundus tadpoles (Berger, 1977; Hotz et al., 1999), or a general selection against parental genotypes (Reyer, Arioli-Jakob & Arioli, 2015).

Conclusion

We found diverse pathways of hybridogenetic reproduction in diploid hybrid males from Eastern Ukraine. Investigating gametogenesis, we observed one or another parental genome elimination followed by endoreplication of the remaining genome in diverse germ cell populations. These pathways resulted in the simultaneous formation of gametes with P. ridibundus and P. lessonae genomes in most males. We found these males crucial for the hybrid’s persistence in these populations because they are the only ones able to form P. lessonae gametes. However, genome elimination and endoreplication have not always occurred correctly, resulting in aneuploidy and the abruption of meiosis in some spermatocytes. We find the gametogenic diversity as the key evolutionary force producing a variety of gametes with different genome compositions and ploidy levels, maintaining these populations in particular and increasing global vertebrate diversity in general.

Supplemental Information

Figure S1 The scheme of sampling localities in the Siverskyi Donets river basin in Eastern Ukraine

Click here for additional data file.

Figure S2 Identification of ploidy level and genome composition of gonocytes, spermatocytes, and spermatids from P. esculentus males collected in the Iskiv pond basin

FISH with RrS1 probe helped distinguish pericentromeric regions only of P. ridibundus chromosomes (indicated by thin arrows). Interphase cell nuclei (indicated by thick arrows) with haploid P. ridibundus (A-C, E, F, I, J) or P. lessonae (A, G, J) chromosomal sets. Mitotic metaphases with 26 chromosomes each of P. ridibundus and P. lessonae (B, E) and with only P. ridibundus chromosomes (J). Meiotic metaphase II with 13 univalents of P. lessonae (A, H) and 13 univalents of P. ridibundus (G). Meiotic metaphase I with 13 bivalents of P. ridibundus (C, D, F, K, L). Spermatids (indicated by arrowheads) with haploid sets of P. ridibundus chromosomes (F-H) and P. lessonae (B, I, L). Scale bar = 10µm

Click here for additional data file.

Figure S3 The overall number of normal and aneuploid chromosomal plates during meiosis (A) and mitosis (B)

The proportion of meiotic plates with different genomes from hybrid frogs collected from the R-E system of the Mozh river (left column) and Iskiv pond (right column). R –genome of P. ridibundus, L –genome of P. lessonae; aneuploidy –number of chromosomes more or less 13 bivalents or univalents.

Click here for additional data file.

Table S1 Number of analyzed spermatids, interphase nuclei, mitotic and meiotic chromosomal plates of P. ridibundus (labeled with RrS1) and P. lessonae genomes in testes from all analyzed frogs

Normal chromosomal plates –prophase I meiosis (13 bivalents), prophase II meiosis (13 univalents); mitotic chromosomal plates—metaphase (2n = 26 chromosomes). Aneuploid chromosomal plates –number of chromosomes more or less than 13 bivalents or univalents for meiosis more or less than 26 chromosomes for mitosis

Click here for additional data file.

Table S2 List and metrics of studied diploid male water frogs Pelophylax esculentus

SVL –snout-vent length

Click here for additional data file.

Supplemental Information 1 Author Checklist - Full

Click here for additional data file.

The authors thank Olexii Korshunov for his help in frog collecting and Olha Biriuk for providing critical comments during the preliminary manuscript preparation, Anna Fedorova for her support and help at different stages of work. We are also grateful to the stuff of the Laboratory of Amphibian Population Ecology and students of VN Karazin National University who helped with animal care.

Additional Information and Declarations

Competing Interests

Author Contributions

Animal Ethics

Data Availability

The authors declare there are no competing interests.

Eleonora Pustovalova conceived and designed the experiments, performed the experiments, analyzed the data, prepared figures and/or tables, and approved the final draft.

Lukaš Choleva conceived and designed the experiments, authored or reviewed drafts of the article, and approved the final draft.

Dmytro Shabanov conceived and designed the experiments, authored or reviewed drafts of the article, and approved the final draft.

Dmitrij Dedukh conceived and designed the experiments, performed the experiments, prepared figures and/or tables, and approved the final draft.

The following information was supplied relating to ethical approvals (i.e., approving body and any reference numbers):

The Committee on Bioethics of the V. N. Karazin Kharkiv National University (minutes 4, 21.04.2016) approved the study.

The following information was supplied regarding data availability:

The raw data are available in the Supplemental Files.

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
