# Peer review of "The high diversity of gametogenic pathways in amphispermic water frog hybrids from Eastern Ukraine"

_PeerJ, doi:10.7717/peerj.13957_

## Round 0.1 · original submission · Minor Revisions

Dear Dr. Pustovalova,

In view of your manuscript, I would like to rectify my decision as per Reviewer 1. Please review the suggestions and resubmit listing all items reviewed.

·

Basic reporting

The quality of English in the manuscript is decent, but needs a thorough read to remove/rectify uncertainties I've spotted (see below). References in the introduciton are given somewhat strange, see my suggestions. The cytogenetic figures are of excellent quality, yet the Suppl. S3(c) - change "R - meiotic metaphase plates with P. lessonae genome" is apparently a typo, change accordingly.

Experimental design

no comment

Validity of the findings

no comment

Additional comments

Review on The high diversity of gametogenic pathways in amphispermic water frog hybrids from Eastern Ukraine.

Line 38. Change the reference “Tunner, 1973” into “Tunner, 1974”.
In his paper from 1973, Tunner described the hybrid albumin phenotype of the Polish P. esculentus complex, showed the hybrid origin of P. esculentus from the P. ridibundus x P. lessonae crosses, and suggested the nonrandom segregation of chromosomes from the hybrid backcrosses of P. esculentus x P. lessonae. While in his 1974 work, Tunner explicitly showed that the inheritance pattern observed in Austrian P. esculentus is the hybridogenesis sensu Schultz, 1969. Graf and Müller (1979) using experimental gynogenesis and producing haploid embryos were the first to confirm conclusions by Tunner (1974), based on the interpretation of allozyme genotyping of artificial crosses, on the hybridogenetic nature of the inheritance pattern in Pelophylax.
Line 41. Remove Tunner, 1973 – in that paper he did not conclude on the ploidy of hybrids, he only suggested “gene-dosage” in some esculentus phenotypes… Check references in Graf and Polls-Pelaz under section “Triploids”, p. 292, and apply accordingly here.
Line 41. Remove reference Graf & Polls-Pelaz, 1989 - it is a review, not an original study.
Line 41. Studies of Berger 1968, and 1971 did not deal with a ploidy of the hybrids… remove the references
Lines 41-44. The references for this sentence are also strange. As late as the work of Tunner & Heppich-Tunner (1991) there were no detailed cytogenetic studies on the mechanism of hybridogenesis in Pelophylax… It is not even cited here.
Line 44. Dolezallkova-Kastankova et al., 2021 – I guess this work has two first authors with shared co-authorship, suggested to be changed to the correct form.
Line 45. Why don't you cite Ogielska, 1994, and most importantly Dedukh et al., 2020?
Whole introduction. In several instances authors cite two old reviews across the text even when they refer to the original studies - Graf & Polls-Pelaz, 1989 and Plötner, 2005, but seems like the most recent review by Dufresnes & Mazepa, 2020 is not familiar to the authors, strange.
Lines 88-90. Please change the coordinates to decimal format
Line 94. What is this “minutes №4”?
Line 99. What does mean “The previous species”
Line 104. In the PDF there is a strange sign before 0.05% - perhaps a matter of wrong formatting
Line 140. “The two geographically isolated populations”, but in the lines 87-90 you have introduced three study sites? The Udy River populations has just disappeared from the study…
Line 153-154. “Nuclei with 5-13 signals contained at least a haploid set of P. ridibundus chromosomes.” Why there were instances with only 5 signals, is your staining is not that specific or were there chromosomes that were not bearing the signal? Is it normal practice to consider the nuclei that have way less than 13 signals than expected from the haploid set of R-genome?
Line 172. “Fifty-four examined interphase cells (n=54)”. Change n=54 to %..
Line 173-174. “The analysis of 14 mitotic chromosomal plates showed 8 plates with 26 chromosomes, of which 13 belonged to P. ridibundus and 13 to P. lessonae.” And the rest 6 plates of these 14, what is their fate?
Line 182. “In one individual (17T-8), we observed interphase nuclei with 3-26 signals (Fig. 2A, B, D) suggesting the presence of haploid P. ridibundus and diploid P. ridibundus genomes in different germ cell populations.” Perhaps needs rephrasing, I have trouble understanding this sentence: 3-26 signals, the haploid set is 13, is it a typo?
Line 184. “14 mitotic chromosomes” – did you mean 14 mitotic chromosomal plates?
Line 230. “from 0 to 14”, again, I am afraid this needs explanation. No signal – L-genome, 13 signals – R genome, why the max is 14 signals. Are there spermatids with 1, 2, 3 etc signals? Won’t they rather be considered sperms that bear 1,2,3 etc R-chromosomes and the rest – L?
Line 261. Add Tunner & Heppich-Tunner, 1991 reference please.
Lines 342-344. Why do you cite Plötner, 2005 – it is a review, not the original study
Line 345. “p=0.000”…
Line 347 – see comment to line 44
Line 358 – same
Line 359. Strange sentence: “However, the well-documented persistence of diploid hybrid males in high abundance over decades of observation in mixed populations of P. ridibundus remains unclear”: what does remain unclear, the documented fact of persistence? I guess needs rephrasing.
Line 367-368. “We hypothesize that the actual prosperity of hybrids may be explained by the hybrid heterosis effect over P. ridibundus (Berger, 1977; Hotz et al., 1999).”
Perhaps better to remove it, it is a very strange statement without any reasoning behind.
Line 383. Change “workers” into “technicians” or “stuff”?
References. I see that your reference list needs additional formatting, e.g. Berger 1977 is followed by Berger 1971 – please check.

Reviewer 2 ·

Basic reporting

The study addresses relevant aspects of gametogenesis in hybrid frogs. I believe that the working hypothesis needs to be clarified to understand the objectives described.

Experimental design

The experiment was well designed. I think the number of animals is small, but considering that the analyzes were performed on the cells, a sufficient number of nuclei were analyzed.
Lines 92-95 lack the collection permission number and ethics committee approval.

Validity of the findings

The results are well described and provide relevant scientific information. The conclusions are in accordance with the proposed objective.

---

## Round 0.2 · accepted · Accept

Dear authors,

The revisions suggested by the reviewers were accepted and corrected in the manuscript. Therefore, I consider it accepted for publication. Congratulations to the study team.